

# Mental health status and quality of life in elderly patients with coronary heart disease

Min Tang[1,*], Song-Hao Wang[1,*], Hui-Lin Li[1], Han Chen[1], Xin-Yi Sun[1], Wei-Wei Bian[2], Jing Sheng[1] and Shao-Jun Ma[1]

[1] Department of Geriatrics, Shanghai Ninth People's Hospital, Shanghai Jiao Tong University School of Medicine, Shanghai, China
[2] Department of Plastic and Reconstructive Surgery, Shanghai Ninth People's Hospital, Shanghai Jiao Tong University School of Medicine, Shanghai, China
* These authors contributed equally to this work.

## ABSTRACT

**Background**. Coronary heart disease (CHD) is the leading cause of morbidity and mortality among elderly individuals. Patients with CHD are at high risk for mental health disorders, and psychological issues may affect the quality of life (QoL) of these patients. Nevertheless, there is little evidence regarding the psychological issues and QoL of patients with CHD among the elderly population. This study aimed to investigate the relationship between comorbidities and mental status as well as QoL among elderly patients with CHD.

**Methods**. Overall, 216 patients were included in this cross-sectional, observational, single-center study. The demographics and clinical manifestations of the patients were collected from electronic medical records. All patients were interviewed using the Chinese version of Symptom Checklist 90 (SCL-90) to assess the mental health status and the World Health Organization Quality of Life-BREF questionnaire (WHOQOL-BREF) to assess the QoL.

**Results**. In total, 96 men and 120 women, with a mean age of 71.69 ± 8.30 years, were included. When controlling for the patients' sex, marital status and stroke, multiple stepwise linear regression analyses suggested that for CHD patients, comorbid type 2 diabetes mellitus had the significant influence on average positive factors (Coef., 5.809; 95% CI [2.467–9.152] $p = 0.001$); when controlling for the patients' sex, marital status and type 2 diabetes mellitus, multiple stepwise linear regression analyses suggested that for CHD patients, comorbid stroke had the significant influence on average positive factors (Coef., 8.680; 95% CI [4.826–12.534]; $p < 0.001$); when controlling for the patients' sex, marital status, type 2 diabetes mellitus and stroke, multiple stepwise linear regression analyses suggested that for CHD patients, comorbid primary hypertension had the significant influence on phobic anxiety (Coef., 0.178; 95% CI [0.010–0.347]; $p = 0.038$).

**Conclusions**. For elderly CHD patients, comorbid type 2 diabetes mellitus and stroke were at risk for psychological problems and lower QoL. Our findings may guide patients and clinicians to make better decisions and achieve better outcomes.

Corresponding authors
Jing Sheng, shengjing60@163.com
Shao-Jun Ma, mashj@163.com

## INTRODUCTION

Coronary heart disease (CHD) is the leading cause of death worldwide (*Wang et al., 2016*). The incidence and prevalence of CHD increase greatly with age (*Kattainen et al., 2006*). Therefore, CHD is the major cause of morbidity and mortality among elderly individuals, which contributes to a substantial economic burden on the healthcare system (*Hanna & Wenger, 2005*). Patients with CHD are at high risk for mental health disorders, such as depression and anxiety (*Bashiri, Aghajani & Masoudi Alavi, 2016*), which have both been strongly related to adverse outcomes in patients with CHD (*Barth, Schumacher & Herrmann-Lingen, 2004*). These psychological problems may not only increase the use of healthcare services, but they also result in disease deterioration (*Dehdari et al., 2009*). In addition, CHD had a severe impact on the quality of life (QoL) (*Dyer et al., 2010*). A previous study has shown that most of the elderly population has three or more comorbid illnesses (*Caughey et al., 2008*). As people age, the risk of chronic conditions increases, such as diabetes, heart disease, cancer and arthritis (*Wu et al., 2013*). Previous research suggested a higher incidence of coexisting diseases in elderly patients, and cardiovascular comorbidities and metabolic diseases were commonly encountered diseases (*Man in't Veld, 1998*; *Van den Akker et al., 1998*). Comorbidity is related to a decline in many health outcomes including QoL and psychological distress, which increases the use of health care resources (*Fortin et al., 2006a*; *Fortin et al., 2006b*; *Wolff, Starfield & Anderson, 2002*). However, until now, there have been few studies that investigated the psychological issues and QoL of elderly CHD patients, and the psychological states and QoL of elderly CHD patients are not clear. Therefore, correlational studies are needed.

The SCL-90 is a widely used, self-rating questionnaire that contains multidimensional subscales to measure overall health status. This scale contains 90 items, using 10 factors, to evaluate the psychiatric symptoms in 10 aspects, which is regarded as a widely used psychological measurement in China and several studies have assessed the validity of this instrument (*Zhao, Feng & Yang, 2019*). The SCL-90 contains a very broad range of psychiatric symptomatology involving feelings, emotion, thinking, consciousness, behavior, lifestyle, interpersonal relationships, diet and sleep (*Wei et al., 2018*).

QoL is an increasingly important health issue in healthcare intervention (*Al-Taie et al., 2019*). The WHOQOL-BREF is a short version of the WHOQOL-100, which is a rapid tool for the assessment of QoL in clinical studies (*Skevington et al., 2004*). The properties of this questionnaire have been tested and verified for evaluation in diverse cultural groups (*Skevington et al., 2004*), which has suggested that it is a valid assessment with higher cross-cultural applicability and it is widely recognized.

Patient self-rating health status measurements may be a useful tool to screen high-risk patients to target for more efficient preventative measures, interventions and treatments in clinical practice (*Spertus et al., 2002*). Due to the adequate reliability and validity of above two scales, we used a Chinese version of SCL-90 to assess the mental health status and the WHOQOL-BREF questionnaire to assess the QoL in patients with CHD, and we conducted a cross-sectional study to investigate the relationship between the comorbidities and mental health status as well as QoL among elderly patients with CHD.

## MATERIALS & METHODS

### Study Design

This cross-sectional, observational, single-center study was designed to investigate the association between the type of comorbidities and mental status as well as QoL among elderly patients with CHD. This study was approved by the China Ethics Committee of Registering Clinical Trials (number, ChiECRCT-2017046) and registered with the Chinese Clinical Trial Registry (number, ChiCTR1900021276). Informed consent was obtained from all patients.

### Study setting and subjects

All subjects of this study were screened as inpatients who were hospitalized in the Department of Cardiology and Geriatrics at Shanghai Ninth People's Hospital, Shanghai Jiao Tong University School of Medicine, between February 2017 and January 2018. A total of 216 patients with a prior diagnosis of CHD (International Classification of Diseases 10: codes I20–I25) were enrolled. The inclusion criteria were those aged ≥60 years, with a prior diagnosis of CHD. The exclusion criteria were patients aged under 60 years, those with organic psychosis, those who refused to give informed consent and who were unable to complete the scale. When patients of the study were hospitalized more than once during the study period, we only used the first record of that time to avoid double counting.

### Procedures and assessment methods

The purpose of the cross-sectional trial was to assess the mental health status and QoL in elderly patients with CHD. A face-to-face interview was performed for each patient in a private meeting room. All interviews were made by the same doctor who received related training. All patients were interviewed using the Chinese version of SCL-90 to assess the mental health status, and the WHOQOL-BREF questionnaire assessed QoL. The demographic characteristics and clinical manifestations of the patients, such as age, gender, height, weight, body mass index (BMI), marital status and concomitant diseases, were collected from electronic medical records, and these characteristics were verified by the patients during the interview.

SCL-90 was comprised of 90 items which consisted of 10 factors, including somatization (12 items), obsessive-compulsive (10 items), interpersonal sensitivity (9 items), depression (13 items), anxiety (10 items), hostility (6 items), phobic anxiety (7 items), paranoid ideation (6 items), psychoticism (10 items) and other factors (sleep and diet, 7 items). Each item was scored using a 5-point scale (0 = none, 1 = slight, 2 = mild, 3 = moderate, 4 = severe). This questionnaire indicated a positive item was with a score of more than 2, which may suggest potential mental health issues for that aspect. A higher score of average positive factors indicated a more severe psychological problem (*Wei et al., 2018*).

The WHOQOL-BREF is a widely used tool that has adequate reliability and validity to assess the QoL. The WHOQOL-BREF questionnaire has 26 items including two general items and 24 items, and 24 items to evaluate four main domains of the QoL, which referred to the physical status(7 items), psychological health(6 items), social relations(3 items), and environmental factors(8 items) (*Ohaeri & Awadalla, 2009*). Each item is scored using a

five-point scale ranging from 1 (very dissatisfied or very poor) to 5 (very satisfied or very good). Mean scores of items are multiplied by four to make domain scores ranging from 4 to 20. And scores of the two general items also range from 4 to 20, respectively.

## Study variables

The parameters, including demographic characteristics, clinical manifestations, concomitant comorbidities and the scores of the two assessment scales, were evaluated in the 216 patients with CHD. CHD with comorbidity of primary hypertension, type 2 diabetes mellitus and stroke, respectively, was analyzed. A set of the following clinical parameters was analyzed: age, sex, height, weight, BMI, marital status, presence of primary hypertension, presence of type 2 diabetes mellitus, presence of stroke, average positive factors and nine factors of SCL-90 scale, the scores of four domains of the WHOQOL-BREF scale.

## Statistical analysis

Data were shown as number (%) or as mean (standard deviation). The t test was used for the continuous variables. The associations were examined using multiple stepwise linear regression analyses, and variables including age, sex, BMI, marital status, primary hypertension, type 2 diabetes mellitus and stroke were entered into the multiple stepwise linear regression. We used multiple stepwise linear regression analyses to generate significant fit models. The coefficients and 95% confidence interval (CI) for each significant variable were determined. A $p$-value of $< 0.05$ was considered statistically significant. Statistical analyses were performed using IBM SPSS 24.0.

## RESULTS

In total, 96 men and 120 women, with a mean age of 71.69(8.30) years, were included. Among the 216 patients, 209 (96.8%) patients were married, and 7 (3.2%) patients were widowed or bachelors. Patients had a mean height of 165.80(8.27) cm, a mean weight of 70.78(9.01) kg, and a mean BMI of 25.71(2.32) kg/m$^2$. The demographic characteristics of elderly patients with CHD are presented in Table 1.

In univariate analysis, the mean scores of average positive factors were 26.18 (14.88) and 20.69 (9.54) for participants with and without primary hypertension, respectively ($p = 0.003$); the mean scores of WHOQOL-BREF physical domain were 12.43 (2.82) and 13.58 (2.08) for participants with and without primary hypertension, respectively ($p = 0.003$). The mean scores of average positive factors were 29.54 (14.97) and 22.23 (12.79) for participants with and without type 2 diabetes mellitus, respectively ($p < 0.001$); the mean scores of WHOQOL-BREF physical domain were 11.74 (2.90) and 13.25 (2.44) for participants with and without type 2 diabetes mellitus, respectively ($p < 0.001$); the mean scores of WHOQOL-BREF psychological domain were 12.93 (2.42) and 13.85 (2.22) for participants with and without type 2 diabetes mellitus, respectively ($p = 0.005$ ). The mean scores of average positive factors were 33.24 (15.66) and 22.57 (12.63) for participants with and without stroke, respectively ($p < 0.001$); the mean scores of WHOQOL-BREF physical domain were 11.31 (2.96) and 13.08 (2.51) for participants with and without stroke, respectively ($p < 0.001$); the mean scores of WHOQOL-BREF psychological domain were

**Table 1 Demographic characteristics of the elderly patients with CHD($N = 216$).**

| Demographic characteristic | Number |
|---|---|
| **Age** (years), **mean** (SD) | 71.69 (8.30) |
| **Height** (cm), **mean** (SD) | 165.80 (8.27) |
| **Weight** (kg), **mean** (SD) | 70.78 (9.01) |
| **BMI** (kg/m$^2$), **mean** (SD) | 25.71 (2.32) |
| **Gender** | |
| Male, N (%) | 96 (44.4) |
| Female, N (%) | 120 (55.6) |
| **Marital status** | |
| Married, N (%) | 209 (96.8) |
| Widowed or bachelor, N (%) | 7 (3.2) |

**Notes.**

Abbreviation: N, number.

12.84 (2.72) and 13.70 (2.18) for participants with and without stroke, respectively ($p = 0.022$). The results of univariate analyses are shown in Tables S1–S3.

Multiple stepwise linear regression analyses suggested that for CHD patients, when controlling for the patients' sex, marital status and stroke, comorbid type 2 diabetes mellitus had the significant influence on average positive factors (Coef., 5.809; 95% CI [2.467–9.152]; $p = 0.001$), interpersonal sensitivity (Coef., 0.151; 95% CI [0.033–0.270]; $p = 0.012$), depression (Coef., 0.241; 95% CI [0.100–0.381]; $p = 0.001$), anxiety (Coef., 0.151; 95% CI [0.035–0.268]; $p = 0.011$), psychoticism (Coef., 0.146; 95% CI [0.045–0.247]; $p = 0.005$), physical domain (Coef., −1.272; 95% CI [−1.953−−0.591]; $p < 0.001$); when controlling for the patients' sex and stroke, comorbid type 2 diabetes mellitus had the significant influence on somatization (Coef., 0.346; 95% CI [0.159–0.533]; $p < 0.001$); when controlling for the patients' age, marital status and stroke, comorbid type 2 diabetes mellitus had the significant influence on hostility (Coef., 0.128; 95% CI [0.015–0.242]; $p = 0.027$); when controlling for the patients' sex, marital status, stroke and primary hypertension, comorbid type 2 diabetes mellitus had the significant influence on phobic anxiety (Coef., 0.165; 95% CI [0.025–0.305]; $p = 0.021$); when controlling for the patients' marital status and stroke, comorbid type 2 diabetes mellitus had the significant influence on paranoid ideation (Coef., 0.097; 95% CI [0.003–0.192]; $p = 0.044$); when controlling for the patients' sex and marital status, comorbid type 2 diabetes mellitus had the significant influence on psychological domain (Coef., −0.855; 95% CI: −1.477, −0.234; $p = 0.007$).

Multiple stepwise linear regression analyses suggested that for CHD patients, when controlling for the patients' sex, marital status and type 2 diabetes mellitus, comorbid stroke had the significant influence on average positive factors (Coef., 8.680; 95% CI [4.826–12.534]; $p < 0.001$), interpersonal sensitivity (Coef., 0.232; 95% CI [0.096–0.368]; $p = 0.001$), depression (Coef., 0.401; 95% CI [0.239–0.563]; $p < 0.001$), anxiety (Coef., 0.143; 95% CI [0.009–0.278]; $p = 0.037$), psychoticism (Coef., 0.305; 95% CI [0.189–0.422]; $p < 0.001$), physical domain (Coef., -1.416; 95% CI: −2.202, −0.631; $p < 0.001$); when controlling for the patients' sex and type 2 diabetes mellitus, comorbid stroke had the significant influence on somatization (Coef., 0.356; 95% CI [0.141–0.572]; $p = 0.001$);

when controlling for the patients' sex and marital status, comorbid stroke had the significant influence on obsessive-compulsive (Coef., 0.308; 95% CI [0.162–0.453]; $p$ <0.001); when controlling for the patients' age, marital status and type 2 diabetes mellitus, comorbid stroke had the significant influence on hostility (Coef., 0.168; 95% CI [0.032–0.304]; $p$ = 0.016); when controlling for the patients' sex, marital status, type 2 diabetes mellitus and primary hypertension, comorbid stroke had the significant influence on phobic anxiety (Coef., 0.320; 95% CI [0.162–0.479]; $p < 0.001$); when controlling for the patients' marital status and type 2 diabetes mellitus, comorbid stroke had the significant influence on paranoid ideation (Coef., 0.214; 95% CI [0.105–0.323]; $p < 0.001$);

When controlling for the patients' sex, marital status, type 2 diabetes mellitus and stroke, multiple stepwise linear regression analyses suggested that for CHD patients, comorbid primary hypertension had the significant influence on phobic anxiety (Coef., 0.178; 95% CI [0.010–0.347]; $p = 0.038$). The relationship between scores of the two scales and three comorbidities is presented in Tables 2 and 3.

## DISCUSSION

Our findings suggested that for CHD patients, comorbid type 2 diabetes mellitus was positively associated with higher scores in average positive factors, somatization, interpersonal sensitivity, depression, anxiety, hostility, phobic anxiety, paranoid ideation and psychoticism, respectively; while comorbid stroke was positively associated with higher scores in average positive factors, somatization, obsessive-compulsive, interpersonal sensitivity, depression, anxiety, hostility, phobic anxiety, paranoid ideation, psychoticism, respectively; comorbid primary hypertension was positively associated with higher scores in phobic anxiety; higher scores represented potential mental health issues. Therefore, the presence of these comorbidities resulted in poorer mental health among elderly patients with CHD. On the other hand, for CHD patients, comorbid type 2 diabetes mellitus was negatively associated with scores of the physical and psychological domain; comorbid stroke was negatively associated with scores of the physical domain. Our results suggested that with comorbidity of these comorbidities resulted in a lower QoL among elderly patients with CHD. To the best of our knowledge, this is the first study to examine the association between the type of comorbidities and mental health status and QoL among elderly patients with CHD.

A previous study reported that co-existing psychological problems were more harmful than a single psychological problem for patients with several basic diseases (*Pincus, Tew & First, 2004*). Multidimensional psychological assessment can aid clinicians to comprehensively and better understand the symptoms and mental health status of patients (*Helzer et al., 2009*). In our study, we used the SCL-90 scale to assess psychological status, which was commonly used in psychiatric and psychological counseling clinics to evaluate the psychological statuses and mental issues among people having different occupations (*Wei et al., 2018*). Several previous studies demonstrated a link between depression and CHD (*Reid, Ski & Thompson, 2013*), diabetes (*Goins et al., 2019*), as well as age (*Reid, Ski & Thompson, 2013*). Besides, mental health disorders are common in patients with

**Table 2** Multiple linear regression analysis of the association between SCL-90 factors and three comorbidities in elderly patients with CHD (*N* = 216).

| Variable | Average positive factors | | Somatization | | Obsessive-compulsive | | Interpersonal sensitivity | | Depression | |
|---|---|---|---|---|---|---|---|---|---|---|
| | Coef. (95% C.I.) | *p value* | Coef. (95% C.I.) | *p value* | Coef. (95% C.I.) | *p value* | Coef. (95% C.I.) | *p value* | Coef. (95% C.I.) | *p value* |
| Sex | 8.355(5.096; 11.614) | <0.001 | 0.418(0.236; 0.601) | <0.001 | 0.275(0.152; 0.399) | <0.001 | 0.162(0.047; 0.278) | 0.006 | 0.243(0.106;0.379) | 0.001 |
| Marital status | −20.357(−29.442; -11.272) | <0.001 | – | – | −0.718(−1.064; −0.373) | <0.001 | −0.482(−0.803; −0.160) | 0.003 | −0.827(−1.208; −0.445) | <0.001 |
| Type 2 DM | 5.809(2.467; 9.152) | 0.001 | 0.346(0.159; 0.533) | <0.001 | – | – | 0.151(0.033; 0.270) | 0.012 | 0.241(0.100; 0.381) | 0.001 |
| Stroke | 8.680(4.826; 12.534) | <0.001 | 0.356(0.141; 0.572) | 0.001 | 0.308(0.162; 0.453) | <0.001 | 0.232(0.096; 0.368) | 0.001 | 0.401(0.239; 0.563) | <0.001 |

| Variable | Anxiety | | Hostility | | Phobic anxiety | | Paranoid ideation | | Psychoticism | |
|---|---|---|---|---|---|---|---|---|---|---|
| | Coef. (95% C.I.) | *p value* | Coef. (95% C.I.) | *p value* | Coef. (95% C.I.) | *p value* | Coef. (95% C.I.) | *p value* | Coef. (95% C.I.) | *p value* |
| Sex | 0.243(0.130; 0.357) | <0.001 | – | – | 0.215(0.083; 0.348) | 0.002 | – | – | 0.181(0.082; 0.279) | <0.001 |
| Marital status | −0.713(−1.030; −0.396) | <0.001 | −0.356(−0.667; −0.045) | 0.025 | −0.512(−0.881; −0.142) | 0.007 | −0.558(−0.816; −0.299) | <0.001 | −0.496(−0.770; −0.222) | <0.001 |
| Age (years) | – | – | −0.011(−0.018; −0.004) | 0.002 | – | – | – | – | – | – |
| Primary hypertension | – | – | – | – | 0.178(0.010; 0.347) | 0.038 | – | – | – | – |
| Type 2 DM | 0.151(0.035; 0.268) | 0.011 | 0.128(0.015; 0.242) | 0.027 | 0.165(0.025; 0.305) | 0.021 | 0.097(0.003; 0.192) | 0.044 | 0.146(0.045; 0.247) | 0.005 |
| Stroke | 0.143(0.009; 0.278) | 0.037 | 0.168(0.032; 0.304) | 0.016 | 0.320(0.162; 0.479) | <0.001 | 0.214(0.105; 0.323) | <0.001 | 0.305(0.189; 0.422) | <0.001 |

**Notes.**

Abbreviation: Coef., Coefficient; C.I., confidence interval; Type 2 DM, type 2 diabetes mellitus.

Reference: Sex-male; Marital status- widowed or bachelor; Primary hypertension-no primary hypertension; Type 2 DM-no type 2 DM; Stroke-no stroke.

Tang et al. (2021), *PeerJ*, DOI 10.7717/peerj.10903

**Table 3** Multiple linear regression analysis of the association between WHOQOL-BREF domains and three comorbidities in elderly patients with CHD ($N = 216$).

| Variable | Reference | Physical | | Psychological | | Social | | Environment | |
|---|---|---|---|---|---|---|---|---|---|
| | | Coef. (95% C.I.) | p value | Coef. (95% C.I.) | p value | Coef. (95% C.I.) | p value | Coef. (95% C.I.) | p value |
| Stroke | No stroke | −1.416(−2.202; −0.631) | <0.001 | – | – | – | – | – | – |
| Type 2 DM | No type 2 DM | −1.272(−1.953; −0.591) | <0.001 | −0.855(−1.477; −0.234) | 0.007 | – | – | – | – |
| Sex | Male | −1.244(−1.908; −0.580) | <0.001 | −0.930(−1.539; −0.321) | 0.003 | – | – | −0.625(−1.091; −0.159) | 0.009 |
| Marital status | Widowed or bachelor | 2.662(0.811; 4.513) | 0.005 | 1.847(0.145; 3.548) | 0.034 | – | – | – | – |

**Notes.**

Abbreviation: Coef., Coefficient; C.I., confidence interval; Type 2 DM, type 2 diabetes mellitus.

stroke (*Almeida & Xiao, 2007*). Similarly, our findings demonstrated that for elderly CHD patients, comorbid type 2 diabetes mellitus or stroke was associated with a higher SCL-90 score; however, further studies with larger samples of patients are necessary to verify our findings. Additionally, in this population, hypertension influenced the factor of phobic anxiety, which may be a useful basis for further research as few studies focus on the comorbidities of elderly patients. Being informed of mental health status might be useful in screening mental health needs, which can guide health care interventions (*Moriarty et al., 2009*). Our study adopted the multidimensional to assess the different aspects of mental health in elderly CHD patients with comorbidities, which may be a useful basis for clinical practice.

CHD had a severe impact on human health and QoL (*Dyer et al., 2010*); therefore, QoL was an important predictor of health outcomes in CHD treatment (*Cepeda-Valery et al., 2011*). CHD patients had a lower QoL, which may be affected by psychological symptoms (*Raymakers et al., 2018*; *Saengsiri, Thanasilp & Preechawong, 2014*). Besides, published studies indicated that comorbidities may affect the elderly patients' QoL (*Jacobs, 2009*; *Rambod, Ghodsbin & Moradi, 2020*), and our study indicated that for elderly CHD patients, comorbid type 2 diabetes mellitus was negatively associated with the scores of the WHOQOL-BREF physical and psychological domains. Therefore, comorbid type 2 diabetes mellitus negatively affected the QoL, which was consistent with previous studies (*Marfella et al., 2018*; *Sardu et al., 2019*). A published study showed that cognitive function was a predictor of the physical function domain of health -related QoL (*Patel et al., 2007*). Also, psychological problems, such as anxiety and depression, are common in stroke patients (*De Wit et al., 2017*; *Froes et al., 2011*). Furthermore, our study indicated that comorbid stroke was negatively associated with the scores of the WHOQOL-BREF physical domain, which was consistent with previous studies. In addition, our study suggested that for elderly CHD patients, comorbid primary hypertension did not significantly influence the scores of WHOQOL-BREF domains. Comorbidity was an important factor in health-related QoL deterioration associated with aging, and it should be noted that comorbidity can affect different domains of QoL to varying degrees (*Zygmuntowicz et al., 2012*). Our findings may serve as a piece of evidence in health care.

There are some limitations to our study. First, our study was limited by its observational nature. The data were from a single hospital, and the sample was relatively small. Second, this was a cross-sectional study in which we identified the relationship between the scores of two scales and the type of comorbidities; however, longitudinal studies are needed to examine the relationship over time (*AlRuthia et al., 2020*). Finally, this is a questionnaire-based survey, and the results might have been negatively affected by acquiescence bias, social desirability bias, and interviewer bias (*Bowling, 2005*). Moreover, there might have been recall bias in elderly patients with a decline in memory. The self-rating questionnaire may result in an additional form of bias (*Murphy et al., 2014*).

## CONCLUSIONS

In this study, we studied the association between the types of comorbidities among elderly patients with CHD and mental health and QoL. The results suggested that for elderly CHD

patients, comorbid type 2 diabetes mellitus were more likely to exist with psychological problems and lower QoL than the population without type 2 diabetes mellitus. In addition, our study suggested that comorbid stroke affected mental health and QoL in the target population. These findings may guide patients and clinicians to make better decisions and help this population to achieve better outcomes.

## ACKNOWLEDGEMENTS

We are grateful to all the participants who took part in this study.

### Funding
The authors received no funding for this work.

### Competing Interests
The authors declare there are no competing interests.

### Author Contributions
- Min Tang and Song-Hao Wang performed the experiments, analyzed the data, prepared figures and/or tables, authored or reviewed drafts of the paper, and approved the final draft.
- Hui-Lin Li, Han Chen, Xin-Yi Sun and Wei-Wei Bian performed the experiments, authored or reviewed drafts of the paper, and approved the final draft.
- Jing Sheng and Shao-Jun Ma conceived and designed the experiments, performed the experiments, analyzed the data, authored or reviewed drafts of the paper, and approved the final draft.

### Human Ethics
The following information was supplied relating to ethical approvals (i.e., approving body and any reference numbers):

This study was approved by the China Ethics Committee of Registering Clinical Trials (number, ChiECRCT-2017046) and registered with the Chinese Clinical Trial Registry (number, ChiCTR1900021276).

### Data Availability
Raw data are available in the Supplemental Files.

### Supplemental Information
Supplemental information for this article can be found online at http://dx.doi.org/10.7717/peerj.10903#supplemental-information.

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
