# Peer review of "Mental health status and quality of life in elderly patients with coronary heart disease"

_PeerJ, doi:10.7717/peerj.10903_

## Round 0.1 · original submission · Major Revisions

Please take into consideration the reviewer’s comments and provide back a point-by-point rebuttal letter addressing those concerns. Their detailed suggestions should improve substantially your manuscript for the second round of review.

·

Basic reporting

English, no substantial problems detected. For some minor comments on language see general comments to the authors below.
References, article structure, self-contained, no problems detected.
Raw data shared, the shared data includes scale scores for SCL-90 and WHOQOL-BREF, not raw data on individual items. And these derived variables are rounded to two decimals. If this is done to protect patient privacy I support the authors' decision on that point (but then this should have been stated). Protection of patient privacy is very important, contemporary malevolent actors may have access to substantial computing resources and sophisticated algorithms, and indirect identification from an open access data file alone is always a threat. I recommend, however, that the authors base their analyses on derived variables with full precision, without rounding.

Experimental design

The design is observational, not experimental. This is not an objection and I regard the article to be within the aims and scope of PeerJ. I regard the design as adequate for the second aim, subject to stated limitations. But since the sample is not clearly recruited to be representative for a specific population I do not consider the design adequate for addressing the first aim. Also, since no analyses described in Statistical analysis or reported in Results are specifically addressing the first aim of assessing the mental health status and QoL among elderly patients with CHD, I recommend that the first aim is deleted.
No other problems regarding the design are detected.

Validity of the findings

I have some recommendations on the description of the instruments, specifically scoring, as detailed in general comments to the authors below. Briefly, the scoring procedures are not precisely described. Data necessary for the analyses reported are provided. Whether data are robust and controlled, I do not have the information needed to assess that. Conclusions are well stated and linked to the second research question, and above I have recommended deleting the first research question. I have, however, some recommendations for revision of the statistical analyses as detailed in general comments to the authors below. I also recommend that syntax for the analyses is included as supplementary information.

Additional comments

1. Main comment on statistical analyses. In unadjusted analyses the authors first test for normality of the scale scores, and since these tests indicate deviations from the normal distribution, a non-parametric procedure is used. In adjusted analysis, however, the authors use linear regression. This is not consistent, linear regression is a parametric procedure. I think that decisions of statistical procedures should not be based on a test for normal distribution. Such tests have low power for small samples, and for large samples normality tests may detect minor deviations with no importance for the validity of linear regression (including t-tests and analysis of variance as special cases). Actually, linear regression has considerable robustness for moderate deviations from a normal distribution, while procedures such as the Mann-Whitney U test are not similarly robust to deviations from its underlying assumption of the same shapes of the distributions within the two samples compared. See e. g. https://bmcmedresmethodol.biomedcentral.com/articles/10.1186/1471-2288-12-78 . Instead of tests for normality I recommend graphical inspections (e. g. by histograms) of the shapes of the distribution of dependent variables, with particular emphasis on the proportions of floor and ceiling (values equal to the lowest and highest possible values. Possible is important here, floor and ceiling are not related to the lowest and highest values in data). If substantial problems with a dependent variable are detected, regression analysis together with a robust procedure for computing standard errors may be considered, such as a bootstrap procedure or Hubert-White estimation. In any case, the results of the normality tests are not reported in the article, they should be reported if these tests are used.

2. Description of scoring of SCL-90 and WHOQOL-BREF. I have made some searches and SCL-90 seems to be a commercial instrument. This may imply that there are restrictions towards making precise statements of the scoring procedures, in case this should be stated. In any case, the lowest and highest possible values (as opposed to lowest and highest actual values in data) for the scale scores should be stated, if possible under the terms for use of the instrument. I have checked with the supplied data, and all scale scores for SCL-90 are between 1 and 5. This may not be consistent with the description of scoring of SCL-90 in the article (line 97-103), where it is stated that the items are dichotomized and the number of positive scores computed. Specifically, the total score for SCL-90 should then be the number of positive items within all 90 items. Also, if this description is correct the scale scores for the domains should be appreciably lower than the total score since the number of positive items are lower, this is not so in data. For WHOQOL-BREF it is presumably possible to state the scoring procedures precisely. In the article it is only stated that each domain has a total rating of 100 points. If this is a statement of maximum possible value, the minimum possible value should also be stated. For WHOQOL-BREF also, this statement does not seem to be consistent with data. Actually, in data the total score for WHOQOL-BREF ranges between 62 and 131, where 131 is appreciably higher than 100. And for the 5 domains the lowest values for scale scores in data are between 4.0 and 6.5 while the maximum values are between 18.9 and 20.0.

3. Other comments on statistical analysis. Since BMI is determined by height and weight, the analyses should not be adjusted for BMI in addition to height and weight.
“The associations were examined using multiple linear regression analyses, and the β estimate and 95% confidence interval (CI) for each significant variable were determined.». This information should be reported also if p>0.05. Variables with p>0.05 should also be included in Tables 2-4. A p-value > 0,05 does not signify «no relationship» (absence of evidence is not evidence of absence). To take one example, the upper confidence bound for Somatization by primary hypertension in adjusted analyses similar to those reported in Table 2 is 0.374, higher than many of the upper confidence bounds reported for SCL-90 domains in Table 2. Therefore, despite a high p-value of 0.232 indicating no strong evidence for this relationship, there is no strong evidence excluding a substantial relationship either. Related to this, there is an ongoing discussion on how p-values should best be used and reported, see e. g. the statement on this issue by the American Statistical Association from 2016 ( http://www.amstat.org/asa/files/pdfs/P-ValueStatement.pdf ), particularly the third principle, «Scientific conclusions and business or policy decisions should not be based only on whether a p-value passes a specific threshold».
The authors may consider including (in Discussion) an assessment of whether the estimated relationships are clinically relevant. The authors have not made formal adjustment for multiple testing, I tend to agree with that since it is not clear within what groups of analyses such adjustment should have been done, but the authors may include (in Discussion) an assessment of whether the large number of estimated relationships might have led to some spurious results. Another less important issue, I recommend replacing «the β estimate» by e. g. «the adjusted difference», this is more interpretable and «β» is often used for standardized regression coefficients that are not of interest here.

4. Discussion and conclusions. There are several statements of associations involving “CHD comorbid with" primary hypertension, type 2 DM and stroke. For readers of the article the context of these statements is clear, but if such statements are cited by others out of context they may be interpreted as associations involving CHD. Such an interpretation is wrong since all patients had CHD. The authors may consider changing the wording of such statements to statements that for CHD patients, comorbid primary hypertension, type 2 DM and stroke were associated with outcomes.

5. Data for analyses have been made available, if possible it had been valuable if syntax for the analyses also had been made available as supplementary material.

6. Details on Results. I have checked the results in the article with the data enclosed. Line 135, mean BMI in my checks is 25.71, not 25.70. Supplementary Table 1, phobic anxiety hypertension mean 1.45, p 0.002, psychoticism not hypertension mean 1.27. Supplementary Table 2, phobic anxiety type 2 DM mean 1.55, standard deviation 0.60. Supplementary Table 3, phobic anxiety stroke mean 1.70, standard deviation 0.60.

7. Other details, notation. language. The notation ± for standard deviations is a bit outdated, use a comma or parentheses instead. Move the software description to the end of Statistical analysis. Inconsistent use of space before references, for example line 31 no space before (Wang et al. 2016), line 41-42 space before (Wu et al. 2013).
Line 52, «which is regarded as the widely used psychological measurement in China». «the most widely used»? In case a reference for this fact had been valuable. The reference to Zhao et al. (2019) is about one specific group only, plateau military personnel». Or «a widely used»?
Line 85, «those with an incompatibility for cognitive disorders», incompatibility with what? Consider rewriting for clarify.
Line 133 and Table 1, change “windowed” to “widowed”.

·

Basic reporting

My mayor concern is related with the statistical analysis.

Experimental design

The design of the study is right. My mayor concern is related with the statistical analysis. Here are my observations:

In the statistical analysis section, it is more correct to mention t tests for independent samples instead of ANOVA, since only two groups are compared.

Regarding the statistical analysis, it is necessary to be more specific and mention the different response variables and the different models generated from multiple linear regression. Important information on the technique used for the construction of the different fitting models is also missing. Finally, everything related to the evaluation of each of the models is also missing, including the evaluation of the different interactions (effect modificatio), collinearity and assumptions of the model (linearity, homoscedasticity and normality). The validity of the models to draw conclusions that are more credible depends on what are indicated above.

Validity of the findings

Results
In Table 1 the recommendation is to put as a footnote an indication that the values are means ± SD for continuous variables or number (%) for categorical variables. You do not need to include the IQR

In Table 2 it is not clear if each variable in column 1 corresponds to separate models or they are in the same model. Nor is it shown if each model was adjusted for other sociodemographic, anthropometric, etc. variables. It is important that it is clear what the response variable was. Let it be clear if the Beta corresponds to the presence and non-presence of each of the studied comorbidities. Same comment for tables 3 and 4.
In Table 1, 2 and 3 of the supplement, it mentions that the Mann-Whitney U test was used, so the most correct is to report the medians instead of the means, since this test compares medians. Tables 1, 2 and 3 only show continuous variables, so it makes no sense to put the n (%) indication in the footnote.

Additional comments

The manuscript is interesting and seeks to evaluate the state of mental health and quality of life, as well as the association between the presence of three comorbidities in patients with CHD with mental health and quality of life.

My final recommendation is to take into account the observations raised

---

## Round 0.2 · Major Revisions

There are pending issues that both reviewers ask you to address. Please provide a comprehensively revised version addressing the editorial comments and a detailed rebuttal letter

·

Basic reporting

No further comments in this second review in addition to general comments to the authors

Experimental design

No further comments in this second review in addition to general comments to the authors

Validity of the findings

No further comments in this second review in addition to general comments to the authors

Additional comments

There are substantial improvements, but I still have some issues. First, the description of the construction of the scale scores for SCL-90 (line 101-113) is as far as I can see unchanged. I have loaded the data fila into SPSS from Excel and checked the ranges of the scale scores for Hostility, Phobicanxiety, Paranoidideation and Psychoticism, both in original and in mean form, and the minimum values of these are greater than zero, in fact the minimum values of the mean scores are always 1. From “Each item was scored using a 5-point scale (0=none, 1=slight, 2=mild, 3=moderate, 4=severe). This questionnaire indicated a positive item was with a score of more than 2, which may suggest potential mental health issues for that aspect. The number of positive items as per the score of each factor and a higher score indicated a more severe psychological problem”, I would have expected that at least some of these mean scores had a minimum value below 1. I still recommend a more detailed and precise description of how these scores are computed. In particular, even if data for individual items cannot be included in the data set supplementing the article for privacy reasons, a decision I support, the authors could still include syntax for the construction of SCL-90 scale scores from individual items (that are stated to range from 0 to 4, line 110) in the syntax and raw output supplement peerj-52224-Data_S2.docx. Concerning WHOQOL-BREF the computation of scale scores is precise now. Still, while it is stated, lines 121-122, that the mean scores are transformed to 0-100 to make interpretation easier, I support this, the scale scores for WHOQOL-BREF in the attached Excel data file range between 4 and 20. It is easy to construct 0-100 scores from this, but these transformed scores are not included in attached data. No reported analyses seem to use the scores transformed to 0-100.
Concerning the regression analyses the authors have expanded regression diagnostics as recommended, if I have understood it correctly, by the other reviewer. I will not recommend reduction of these diagnostics, with one exception (Durbin-Watson as detailed below). But I still recommend re-running at least some of the regression analyses with estimation of standard errors that is not dependent of the standard assumptions underlying least squares estimation. All the common possible deviations from these assumptions, including normality of error terms and homoskedasticity (or homoscedasticity, the spelling is disputed), have at least three issues in common; they do not bias the estimated regression coefficients, they may bias standard errors, usually downwards, and bootstrap estimation procedures do not assume that these assumptions are satisfied and give valid inference even if some of these assumptions are violated. For the convenience of the authors I attach syntax for the analyses (without some diagnostics for simplicity), including a bootstrap setup for the first regression analysis. The authors may consider running the regression analyses with bootstrapping, and report confidence intervals and p-values based on these if there are substantial deviations.
Concerning the Durbin-Watson test this is a procedure mainly to check for autocorrelation in time series. It may be possible to extend the procedure to observations with some other meaningful ordering, such as spatial data, but a Durbin-Watson test for observations with no statement of an interpretable ordering does not make sense. If the order of the respondents (rows in the data file) is changed, the Durbin-Watson statistics will change.
I have run the analyses reported in Tables 1-3 based on the supplementary Excel file imported into SPSS, and detected some minor differences in Table 2. And in table 3 most upper confidence bounds are lacking. I have also checked the numbers reported in Results.
Details on regression analyses, Table 2. To get anything close to Table 2 the mean versions of scores have to be used. Somatization, primary hypertension, p-value 0.854 in Table 2 vs 0.855 by running from syntax. Type 2 DM, coefficient 0.355 vs 0.354. Stroke coefficient 0.371 vs 0.370. Interpersonal sensitivity, primary hypertension, p-value 0.108 vs 0.107. Type 2 DM, p-value 0.042 vs 0.043. Stroke lower confidence bound 0.090 vs 0.089. Depression, primary hypertension coefficient 0.131 vs 0.132, lower bound -0.045 vs -0.044, upper bound 0.307 vs 0.308, p-value 0.143 vs 0.141. Type 2 DM, upper bound 0.354 vs 0.353. , Hostility, , primary hypertension coefficient 0.062 vs 0.061, Type 2 DM, coefficient 0.111 vs 0.112, Stroke lower bound 0.016 vs 0.015, upper bound 0.291 vs 0.290. Phobic anxiety, primary hypertension, lower bound 0.030 vs 0.031, upper bound 0.371 vs 0.372, Type 2 DM, lower bound 0.016 vs 0.015, stroke coefficient 0.280 vs 0.279, lower bound 0.117 vs 0.116. Paranoid ideation, lower bound -0.075 vs 0.076, stroke upper bound 0.339 vs 0.338. For Table 3 I have not found any discrepancies, but as commented above most upper confidence bounds have been deleted.

Details
Line 21, “was associated”, change to “were associated”, diabetes mellitus and stroke together constitute grammatical plural.
Line 27, “had no association”, as detailed in my first review there is no such thing as “no association”. If this statement is based on a pa-value > 0.05 state that, if the estimated association and both its confidence bounds support an interpretation as no substantial association state that.
Line 122, consider changing to e. g. “to ease the interpretation of the four domains”. In any case, “easily” with the ending “-ly” is not grammatically correct since this characterizes “interpretation”, a noun.
Line 176, “The results of univariate analyses are shown in Supplementary Tables 1, 2, and 3.” Table 1 is descriptive and Tables 2-3 report from regression analyses with more than one independent variable, none of these analyses are univariate.

·

Basic reporting

In the abstract and particularly in the explanation of the models, it is necessary to include that it was adjusted for BMI and hypertension and for all the variables in the model. The negative association also has to be explained as an adjusted association and adding the coefficient values, and p-value. Confidence interval not necessary.
In the statistical section I suggest you use STEPWISE to generate significant fit models. Add a paragraph on how each of the models will be evaluated.
In table 1, 2 and 3, I suggest to delete column for statists t, since it is not necessary.
In the paragraph of lines 177 to 183. It is not correct to interpret that there is an association of variables when the value of p is not significant. I think the association models are the main part of the manuscript and it should explain each of the models, including the coefficient values and p-values, and mention all the adjustment variables that have remained in the model, for example if in the model remain the variable hypertension and diabetes, then mention that each of them was adjusted by each other. It should include a main table where all the models with the coefficients, confidence intervals and p value are included. As a footer put the adjustment variables.

Experimental design

No comment

Validity of the findings

Review my comment

Additional comments

In the abstract and particularly in the explanation of the models, it is necessary to include that it was adjusted for BMI and hypertension and for all the variables in the model. The negative association also has to be explained as an adjusted association and adding the coefficient values, and p-value. Confidence interval not necessary.
In the statistical section I suggest you use STEPWISE to generate significant fit models. Add a paragraph on how each of the models will be evaluated.
In table 1, 2 and 3, I suggest to delete column for statists t, since it is not necessary.
In the paragraph of lines 177 to 183. It is not correct to interpret that there is an association of variables when the value of p is not significant. I think the association models are the main part of the manuscript and it should explain each of the models, including the coefficient values and p-values, and mention all the adjustment variables that have remained in the model, for example if in the model remain the variable hypertension and diabetes, then mention that each of them was adjusted by each other. It should include a main table where all the models with the coefficients, confidence intervals and p value are included. As a footer put the adjustment variables.

---

## Round 0.3 · Minor Revisions

Please take into consideration the reviewer’s comments and provide a revised version addressing those concerns.

·

Basic reporting

No further comments in this third review in addition to general comments to the authors.

Experimental design

No further comments in this third review in addition to general comments to the authors.

Validity of the findings

I have one important comment regarding the use of stepwise regression, details are given in general comments for the authors below.

Additional comments

I have only one important issue now, the use of stepwise regression, due to a recommendation from the other reviewer in the second review round. While it is understandable that the authors followed this recommendation I have strong reservations against stepwise regression and therefore cannot recommend publication as long as this procedure is used. Since this is a case of fundamental disagreement between two reviewers I think the issue has to be resolved by the Editor. In any case this recommentation is as far as I can see not related to any recommentations from the other reviewer in the first round. In contrast, my recommentations in the second round are mostly related to changes from the first to the second versions of the article, or questions for further clarifications.
Concerning stepwise regression, I follow the recommendations of Frank Harrell’s book on regression modeling strategies, https://link.springer.com/book/10.1007/978-3-319-19425-7 , in particular the section on variable selection in chapter 4, probably section 4.3. Due to the pandemic I have not at present access to my office where I have this book, but I think the information above is correct. Briefly, independent variables that are selected by a stepwise procedure have p-values , standard errors and lengths of confidence intervals biased downwards. Also as shown in a large number of simulation studies the set of independent variables surviving a stepwise procedure is unstable, particularly for small and medium size data. Open access sources making the same point are numerous, to give a few, https://towardsdatascience.com/stopping-stepwise-why-stepwise-selection-is-bad-and-what-you-should-use-instead-90818b3f52df , https://journalofbigdata.springeropen.com/articles/10.1186/s40537-018-0143-6 , http://www.danielezrajohnson.com/stepwise.pdf , http://assodiaf.org/en/how-bad-is-stepwise-regression/ .

·

Basic reporting

See comments in General comments for the author

Experimental design

No comments

Validity of the findings

View comments in General comments for the author

Additional comments

In general, improvements are observed in the manuscript, although there are still some observations that could improve the manuscript even more. These observations are described below.

In the paragraph between rows 178-189, in the explanation of the models, it does NOT refer to the fact that each of the coefficients is adjusted by a series of variables. In some models, diabetes was also adjusted for stroke and vice versa, in addition to the rest of the variables in the model. The adjustment variables were not common in all the models, so it is not possible to summarize the adjustment variables for all the models; it must be individualized for each model or groups of models. On the other hand, it is important that in the interpretation of the coefficients it refers to the type of positive or negative association (take into account the feasibility of this association), in addition to statistical significance. Apply the same for the corresponding section in the summary. I think that in this same paragraph (178-189) all the models and their results should be referred to or, where appropriate, eliminate them.

In discussion edit line 198
I think that the results of all the models should be discussed, if not eliminate them from the objectives

In the regression related tables, I suggest that the information corresponding to the constant be eliminated. For table 13, put in the footer that no variable was selected in the model and add a hyphen as a value. The latter applies to model 14 and 15 as well.
Table 2. Model 1 does not have age or BMI and in the abstract it is mentioned that models were adjusted by age and, BMI etc.

Variable age in model 7 it is necessary to put the units (years for instance).

In Table 3 is missing the confidence intervals data for model 11. Nor is information presented for model 13.

---

## Round 0.4 · Minor Revisions

Regarding the conflict between the reviewers' comments about the statistical methods, I personally suggest using the stepwise method, although at this point is the author's decision which method they use. Just justify it properly. 

I would not limit the acceptance of the manuscript upon the use of one or another method. It would help if you used what you consider more adequate and justify.

---

## Round 0.5 · accepted · Accept

Thanks for addressing the reviewer's comments. The issues for the statistical analysis are adequately addressed and properly addressed and explained in the final version.